# Fitting Social Enterprise for Sustainable Development in Vietnam

**Minh Hieu Thi Nguyen [1,2]**, **Darrin James Hodgetts [2,]* and Stuart Colin Carr [2]**

1   Faculty of Management and Tourism, Hanoi University, Km9, Nguyen Trai Street, Hanoi 100000, Vietnam; hieuntm@hanu.edu.vn
2   School of Psychology, Massey University (Albany Campus), Private Bag, 102904, Auckland 0632, New Zealand; S.C.Carr@massey.ac.nz
*   Correspondence: D.J.Hodgetts@massey.ac.nz; Tel.: +64-9-414-0800

**Abstract:** Drawing on aspects of both commercial and not-for-profit organisational structures, social enterprises strive to become financially sustainable in order to support efforts to address various societal problems, including poverty and socio-economic exclusions. This study documents the experiences of 20 social entrepreneurs regarding the fit between their leadership practices, social enterprises and the Vietnamese societal ecosystem. Results from semi-structured go-along interviews foreground the importance of fit between the societal eco-system, key cultural values and relational practices, entrepreneur leadership and the structure and functioning of social enterprises in achieving their pro-social missions. This article contributes to emerging literature on the sustainability of social enterprises in emerging economies and is currently being drawn upon in the development of policy responses in Vietnam.

**Keywords:** social entrepreneurship; person–environment fit; leadership; pro-social impacts; culture

## 1. Introduction

Historical developments of social enterprises vary across countries. Such developments have mushroomed from the 1990s, particularly in countries facing rampant inequalities and associated socio-economic difficulties relating to employment and disparities in health, education and gender [1]. There has been a significant rise of social entrepreneurs taking leadership to address such issues and contributing to sustainable growth in many emerging economies [2]. Even within so-called developed economies, including the United States and Europe, the contributions of social enterprises to addressing socio-economic concerns is increasingly acknowledged. For example, in 2020, across European Union member states, social enterprises created 13.6 million jobs [3]. O'Brien [4] and The British Council [5] estimate that 100,000 social enterprises operating in the United Kingdom employ 2 million people; and in the United States, social enterprises are estimated to represent 3.5% of the GDP, which is more than Silicon Valley. In terms of sustainability in the context of the present Covid19 pandemic, a study of 38 countries by The British Council in partnership with Social Enterprise UK and United Nations ESCAP [6] found that social enterprises had proved resilient during the recent pandemic, with only 1% shutting down and a further 7% temporarily closing their services.

Academic publications on social entrepreneurship have increased significantly globally, though research is focused mainly on North America and European contexts, which have produced more than 60% of previous publications [7]. Extant research has a wider reach, however, and encompasses foci on the attributes of social entrepreneurs, organizational characteristics, and relationships between social enterprises and the socio-economic contexts within which they operate [8]. To date, scholars have paid far less attention to these issues in emergent economies. Such a focus is important because in contexts such as Kenya, for example, where social enterprises contribute 45% to the GDP [9]. More



research is also needed to extend present understandings of the role of leadership and cultural considerations regarding the functioning of these enterprises both within and across diverse settings [7,8,10]. Correspondingly, this article considers cultural motivators for the development, leadership, functioning and pro-social missions of social enterprises in Vietnam.

In terms of sustainable development, social entrepreneurship is central to an organizational strategy for addressing societal issues, often poverty and exclusion [11]. These enterprises seek to balance sound commercial and pro-social imperatives [12]. Leadership in these organizations can be challenging [13] and involves pro-social efforts to ensure increased social and economic inclusion [8]. What is clear is that these enterprises reflect the core values, experiences, and knowledge of their founding entrepreneurs [14,15], who offer strategic leadership and generally embrace ethical responses to the needs of a range of stakeholders, rather than just their shareholders [16]. Both the entrepreneurs and their enterprises are immersed within societal and cultural ecosystems that play important roles in shaping what the organizations are seeking to achieve and how they are led to achieving their missions [10,13]. Although emerging, research on the influences of culture and other contextual factors on the leadership of these organizations remains nascent [7,8,17].

This emerging leadership focus speaks to issues of fit between social entrepreneurs, their enterprises and the eco-systems within which they are operating [10,18]. Person–environment fit theory comprises a prominent line of theorizing and inquiry within management and organizational psychology that offers fruitful insights for extending current research agendas on social entrepreneurship [19–21]. This theory foregrounds the importance of person–organization fit [22], for example, how founders often build organizations that reflect the values they embody [13,23]. Scholarship in the area has also extended out to organization-to-organization fit in the context of mergers [24], and investigations of the extent to which organizational cultures fit with broader societal cultural values including leadership [25,26]. Despite these conceptual developments, research that simultaneously investigates how social enterprise founders (leaders), their organizations and societal cultures fit to enable enterprise success or failure remains absent from the scholarly canon.

In the most relevant study we know of, Nguyen, Carr, Hodgetts and Fauchart [10] conducted a quantitative survey of social entrepreneurs in Vietnam, whose responses suggested that the fit between participants and their enterprises, and the broader societal eco-system was significant in terms of organizational performance: Efficiency (revenue, cost, profit, job creation) and generosity (contribution to the society, change in the quality of work-life for employees). Further, the perspectives and decision-making practices of social entrepreneurs were found to result from their distal (e.g., institutional and cultural environment) as well as proximal contexts (e.g., family, personal situation) [17,27,28].

Building on the scholarship of Nguyen, Carr, Hodgetts and Fauchart [10], the present article explores multi-layered forms of fit as these relate to the social enterprise sector, which prioritizes poverty eradication and socio-economic inclusion through the leadership of social enterprises in an emergent economy, Vietnam. In doing so, we draw on and extend existing conceptualizations of fit to consider the dynamics of social entrepreneurship as these relate to the nexus of fit between social entrepreneurs, their enterprises, the broader social cultural context and their pro-social social performance in Vietnam.

Vietnam is an ideal location for exploring these issues. According to recent reports of the British Council, the United Nations Economic and Social Commission for Asia and the Pacific (ESCAP), and Social Enterprise UK in February 2021, 80% of Vietnamese social enterprises reported positive performance achievement and shared 50% of the profits generated with their staff and related beneficiaries. This is the highest rate of such resource transfer in Southeast Asia. In this paper, we propose that these trends reflect long-held cultural values [29] of economic generosity and duty of care towards others that stem from the traditional functioning of Vietnamese villages as well as Buddhist and Confucian belief systems that centralize the interconnection of human beings. We will document how central the harmonious relationships are between traditional village values

of inclusion, care and generosity [30] that fit with social entrepreneurs (leaders), and the social enterprises (organizations) they found.

Organizational literature on Person–Environment fit, conducted largely in W.E.I.R.D (Western, Educated, Industrialized, Rich and Democratic) contexts [31], has resulted in two main types of fit: Supplementary and Complementary. While supplementary fit foregrounds psychological processes of similarity-attraction in terms of values, goals and personality [23], complementary fit instead stresses the importance of how different personalities or a person and an organization may complement one another [32]. Although these two approaches purport to focus on the person–environment fit, they both rather uncritically accept the 'individualism' that pervades WEIRD psychologies today [33], and focus on relationships between such individuals whereby personal needs and interests are prioritized and separated from collective or contextual needs [20].

This classic individualistic focus contrasts with more collectivist-orientated psychologies that operate in contexts such as Vietnam that place more emphasis on mutual responsibilities and interconnected selves [33]. Vietnam has a 4000-year history of unique cultural and socio-economic development that has withstood and adapted to waves of invasion. The country was colonized by China for 1000 years, by France for more than 100 years (until 1954) and reunified after the overthrow of subsequent occupation by the United States [34]. In recent history, the country also experimented with a centrally planned economy, which was followed by an open-door period that led to the present socialist-oriented market economy [35]. Whilst remaining true to its collectivist village values of interconnection and mutual support, the country's culture has also been influenced by an eclectic mix of Buddhist, Confucian and Taoist and Western inflections, including socialism and, later, enterprise capitalism [29].

In surviving millennia of invasion, resistance and renewal, Vietnamese people have come to believe in the need to find balance and harmony in life, even when faced with conflict, hardship and struggle. Despite the hardships collectively remembered, harmony has emerged as the key principle within the country's culture [30], and a valuing and enacting of harmonious connections of "Individual–Family–Village–Nation" underpins the sense of collectivism and mutual responsibilities and obligations [29,34]. Therefore, it is likely that the self-reflections of social entrepreneurs interviewed for the present research will entangle culturally laden understandings of leadership, enterprises and their missions [36,37].

As noted above, a key aspect of Vietnamese society and culture is the 'Village' (Làng)—a key foundation that is embedded deep within the Vietnamese psyche and has enabled Vietnamese families to adapt to and survive various upheavals and challenges [30]. The 'Village' is often referred to as the smallest governmental entity in Vietnam [29]. Each person in Vietnam learns to identify themselves with a village, which helps them to cope with hardship and contribute to something larger than themselves [38]. Correspondingly, we will argue that it is core Village values of Vietnamese people [39,40] that underpin much of the orientation, relational and leadership practices and pro-social impacts of contemporary social enterprises. Central is a valuing of community ties and supports that extend out beyond and connect across different ethnic groups. For instance, a popular proverb is:

"Squash, take care of melon (Bầu ơi thương lấy bí cùng)
Despite your different races, you grow under same roof (Tuy rằng khác giống nhưng chung một giàn)"

This proverb reminds people about the core values of compassion and inclusion in shaping positive and mutually supportive relationships among people in communal settings that extend out beyond familial or clan networks today. Recognition of the need for a sense of unity and efforts to help each other that stems from the traditional village have remained pervasive for some time [39]. We will argue that in establishing social enterprises, entrepreneurs in Vietnam emphasize village and pro-social values of inclusion, care and generosity *through* their strategic leadership. As such, the everyday leadership practices

that underline the functioning of social enterprises in Vietnam are in keeping with i.e., fit and reproduce aspects of a shared cultural heritage that enables them to contribute to the cultivation of more equitable and inclusive socio-economic structures.

To recap, this article documents the Village values and relational practices that shape the worldviews and leadership of social entrepreneurs, and the focus and functioning of the enterprises they have created in Vietnam. We will show how this cultural adaptation of social enterprise leadership to the Vietnamese context also increases the fit between the entrepreneurs, the organizations and the broad socio-economic environment within which they are operating. We are particularly interested in documenting how participating entrepreneurs implicated the dimensions noted above in their efforts to achieve their pro-social missions [41].

## 2. Method

Twenty social enterprise founders participated in the present study. All satisfied the following criteria: *"(1) Social mission is the top priority; (2) using business activities, fair competition as tools to meet social objectives; by (3) re-investing profit generated from business activities into the organization, communities and social objectives"* which were defined by Cung, et al. [42]. All had at least three years' experience in the sector. Thirteen were recruited from our previous quantitative survey Nguyen, Carr, Hodgetts and Fauchart [10] and had chosen the option of participating in a follow up in-depth interview. A further seven participants were recruited from the British Council and Center for Social Innovation Program (CSIP) database of local social enterprises. No further interviews were required as we had reached the point of qualitative saturation and very little new information was being disclosed after the completion of the first 10 interviews. In total, 20 h of interviews were obtained from 12 male and 8 female participants.

The authors designed the interviews to explore the results of a quantitative study by Nguyen, Carr, Hodgetts and Fauchart [10] that focused on links between social entrepreneurs, social enterprise performance and ecosystem supports. The authors designed a semi-structured interview guide containing a list of 30 prompts relating to 6 key issues that were found to be significant in the earlier quantitative study: (1) The history of the enterprises, (2) leadership, (3) the environment, (4) resources, (5) social impacts of social enterprises, and (6) reflection and future. During the interviews, participants were afforded opportunities to reflect on their social entrepreneurship from inception, in relation to their personal backgrounds, leadership orientations, the societal ecosystem in which they operate and the social impacts of their operations. All interviews were semi-structured [43] and engaged the first author and participants in go-along conversations where questions were open ended and the line of inquiry was adapted to the situation as the first author was taken on tours of the organizations [44,45]. As is common practice in semi-structured interviews, prompts were used flexibly and not all questions were asked of all participants or in the same order. Key prompts included: "Please tell me a story about how this enterprise came about?", "How is leadership effective in your enterprise?", "What things help/harm the success of social enterprises?", "Do you think social enterprises are having a positive impact? Do you think some enterprises are having more impact than others?", and "If you went back in time knowing what you know now and were about to start your enterprise again, would you?". Before closing each interview, the first author summarized the main points from the conversation to help spark further dialogue. All interviews were conducted in Vietnamese. A small gift was offered to participants as an appreciation and cultural norm of Vietnamese to break the ice and to engage in open communication [46]. In the interviews, all the participants raised issues around the importance of cultural processes and relationships without pre-planned prompting from the first author. Correspondingly, the focus of our analysis was expanded beyond the set of issues that underpinned the interview prompts. This is in keeping with the abductive approach taken to the analysis [36,47].

Each participant was assigned a pseudonym to be used in the analysis and their recorded interview, which was transcribed into Vietnamese. Preliminary analysis was conducted in Vietnamese with input from the two non-Vietnamese authors through constant dialogue. This iterative process was necessary to preserve the cultural nuances central to participant accounts in Vietnamese whilst allowing us to also explore emerging issues in relation to international theory and research that is published in English [48]. The emergent focus of the analysis arose through a combination of deductive topics that had surfaced from the quantitative survey and related interview prompts and the inductive issues participants raised in the context of our go-along conversations. We then went back and systematically coded all dialogue pertaining to the topics listed in Table 1, which comprise the conceptual framework for the analysis. We then engaged in further dialogue as a team in selecting relevant exemplars for each topic, which was then translated into English for further interpretation. At this point, theoretical concepts and literature about Vietnamese culture were also used as an interpretative resource to provide a conceptual context in unpacking and working through the broader cultural significance of what participants were talking about [49]. All authors then worked together through a process of writing as analysis, to drafting and redrafting the analysis as presented in the following section [36,43,50]. This iterative and cross-cultural strategy of analysis was necessary in order to preserve and unpack important cultural nuances and concepts pertaining to the core focus of the study, and to relate emerging findings to existing theory and research [49].

**Table 1.** Key components in the overall interpretive frame for the analysis.

| Theme | Category | Unit |
|---|---|---|
| Context | Culture | Think about other |
| | | Harmonization |
| | | Mercy |
| | Personal situation | Difficult life experience |
| | | Studious |
| Social entrepreneurs | Resilience | Hardship to maintain both social and business objectives |
| | | Persistently dealing with challenges |
| | | Self-healing |
| | Faith in Fate | Belief in cause-effect |
| | | Supporting others is their life mission |
| | | Trust in the life's beauty |
| | Strategic leadership | Inclusive leadership |
| | | Solidary vision |
| | | Pro-activeness |
| Social enterprise—The village | Devoted to community | Understand the community |
| | | Believe in social mission |
| | | Commit to social impact making |
| | Partnership oriented | Open for collaboration |
| | | Partnership with all stakeholders |
| Pro-social efficiency | Efficiency | Finance achievement |
| | | Income improvement |
| | | Job security for employees |
| | Generosity | Mental and physical health improvement for relevant stakeholders |
| | | Customize benefit for to fit with the needs of disadvantaged groups |
| | | Equity for disadvantaged people |

Central to the resulting analysis are key cultural values and relational practices that appear central to the accounts of participants and the orientations of their social enterprises, including generosity and studiousness of spirit, benevolent leadership in service to others, solidarity and a strong belief in destiny or fate. This belief in fate appears to function to enable participating entrepreneurs to cope with adversity in the form of barriers to progress and failures without becoming despondent. The following analysis explores manifestations of core village values in the formation, orientation and operation of their enterprises. We are particularly interested in how the enactment of these values facilitate the fit between entrepreneurs, their social enterprises and the local village communities that populate the broader societal eco-system. This focus is important because the fit between these elements is fundamental to the strategic leadership of social enterprises in the delivery of pro-social efficiency in addressing pressing social problems [13,37].

## 3. Findings

The analysis is presented in two sections. The first focuses on the cultural background, values and inclusive strategic leadership style that participating entrepreneurs draw upon in creating their enterprises in a manner that fits Vietnam today. The second explores how the strategic leadership and cultural values of participating village-styled enterprises reinforce their pro-social performance.

### 3.1. Fit between Social Entrepreneurs and the Cultural Ecosystem

Participants repeatedly proposed that the primary rationale on which they founded their social enterprises related to fate and destiny. Vietnamese people are socialized to believe that they do not fully control their destinies or lives [51]. This does not mean that they cannot exercise agency over their actions. It does mean that they are attuned to external influences beyond their control and the need to harmonise their efforts in relation to external factors [30]. This cultural mindset is logical when we consider the country's long history of war, invasion and subjugation [34].

Correspondingly, participants emphasized the need to work towards realising one's fate or intended contributions in life, but to not blame oneself if one's aspirations are not fully realized. Participants positioned themselves as part of a larger cosmic flow (life course or karma) that is more harmonious when one is moving with the flow, rather than trying to swim against it. For example, Nhan is a social entrepreneur devoted to traditional handicraft production and promoting disability wellness through job creation, and proposed that she was born for this work:

> Since I finished my high school, I didn't do any other jobs. I followed this job since 18 years old ... Many people quit, but I don't want to. I like the handicraft work. I think this is my karma, my destiny. Now, it has been 38 years that I am doing this job (Nhân).

Such accounts are central to the assertion of a necessary fit between the person and their assigned character or fate as a social entrepreneur. Participants expressed a profound sense of belonging when realising their fate, which helps maintain their efforts to persist in assisting others. Several participants extended such statements to propose that central to their destinies is service to others through pro-social missions. They invoked a cause-and-effect logic in that the condition of fate sets the stage that results in their social entrepreneurship. Several introduced fate as a force that sets the conditions (*duyên*) for success or failure:

> We can't reach our expected goals because we don't have enough condition (đủ duyên). Everything depends on fate, we haven't had the conditions to achieve our goal (Cao).

These extracts reflect aspects of how participants invoke a Viet worldview that centralizes the role of fate in setting their orientations in life [51], and the conditions for the performance of their enterprises. These dynamics of fate and personal agency are well

known through the Viet proverb '*Mưu sự tại Nhân, Hành sự tại Thiên*' (Humans make plans, Heaven makes them fail or succeed) [52].

Such thinking can be liberating for participants in that it harmonises an external and an internal locus of control. This worldview offers a source of purpose and strength in pursuing social missions. Participants are able to go with the flow and accept their roles and responsibilities towards others that come with their fate. Here, fate is not simply an abstract concept. Fate exercises agency in guiding entrepreneurs to the resources and support that they need to achieve their missions. As Thao states:

> I strongly believe that when we really need help, help will come to us. You don't even have to call for help, it will come, surely. I am sure about this. If the help does not come, it meant you don't really need it now. Don't wait, don't expect any help! Concentrate on what you are doing, do things that you are good at.

Here, the belief in fate does not necessarily equate to a passive approach to achieving one's goals or seeking help. Rather, there is a combination of ideas in that one needs to concentrate on what one is doing and to get good at it, and if one does these things, one is more likely to be successful. If things do not work out, then perhaps this is as fate intended [53]. This way of thinking about social enterprises enables participants to share or diffuse responsibility between the force of fate and their own proactive actions within an uncertain societal setting. It enables entrepreneurs to focus on trying to overcome challenges as these arise and to work through problems themselves.

One's own agency does not only shape one's own fate. In acting generously towards others, one is also likely to have such efforts reciprocated [54]. For example, Thuy's mission is to build a happy and effective community of teachers who will then support the education and economic inclusion of students. By following her passion for helping others, Thuy not only contributes to a better world, but will also cultivate a successful life:

> The person who receives enough love, they won't hurt others. If people are all respected, we don't need to work on human rights. You totally can use your passion to help others and earn money. The more you help others, the more money will come to you.

Such extracts reflect the common belief that doing good things returns good fortune [54]. The key cultural assumption here is that if one shares with, and devotes themselves to helping, others to the best of their ability then positive outcomes will come back to oneself. The understanding expressed around the role of fate and pro-social cause-and-effect in life seem to motivate social entrepreneurs. Central to participants' accounts regarding core values is the need to have and exercise a warm-hearted or benevolent form of leadership that in the Confucian system is associated with the character Ren [55,56]. Whilst fate sets the conductions, human beings still need to exercise agency through a studious spirit and benevolent kind-heartedness by dispersing good deeds/seeds to grow or ensure that positive destinies are realized for the self and others [19,51].

Many participants also talked about how difficult life experiences and witnessing hardship offered particular motivations for their social missions. These difficult experiences were often associated with their families:

> Before, my parents fought a lot. I was a special student. I wanted to kill myself many times. Or leave my family. Luckily, there was a good teacher. I love her so much. Because of her, I turned into a good student (Thuy).

Taking inspiration from adversity and in accord with the cultural logic of fate, Thuy works to cultivate a happy teaching community as a way to not only practice her karma, but to contribute to the karma of others. Likewise, Tu also talked about experiencing difficult life events with a mother living with a disability and how this shaped his path towards a related social mission:

> In fact, my mother has a disability. Since I was a child till now, I do have a huge sympathy with disadvantaged communities as a result. My father was a soldier,

we are not rich. When I was 18 -years-old, I got a government scholarship to study overseas. I really appreciated it. It changed my life. Since then, I realized the value of education. If I didn't have that scholarship, my life might be floating somewhere with no reputation. That is why I started my social enterprise in education (Tu).

From his formative experiences, Tu developed social enterprises in education that provide free tuition and job opportunities for disadvantaged young people. Further reflecting the studious spirit, such participants presented adversity as opening up opportunities for growth and actions, rather than being challenges that lead to inaction [30]. Evident in such accounts is how efforts to help others are presented as a logical continuation of a life-course and central to their very sense of self and purpose [54]. Whilst participants emphasized the experiential and social aspects of their efforts, they also acknowledge the importance of commercial considerations in achieving their destinies.

It is crucial to note that hardships are not simply presented as motivating factors from the past. For example, several participants worked fulltime in other jobs and used this income to cover financial shortfalls in the social enterprises they were leading. They talked about the resulting pressures they faced in developing sustainable social enterprises:

I have to cover the cost of the social enterprises by my fulltime salary. I have my social enterprise team meeting at lunch. In the evening, after work, I follow up correspondence, work for my social enterprises (Uu).

Such extracts speak to the depth of commitment these participants embrace in terms of creating opportunities and livelihoods for others. These also reflect the pressures that often come with leadership in this sector, which are met by the studious Vietnamese spirit. The importance of persistence or studiousness of spirit was repeatedly raised throughout the interviewees. These leaders do not regret making considerable sacrifices to support others: *"I don't think I have anything to regret!"* (Cao).

Interviewees reported making a range of sacrifices to support the development of their social enterprises. For example, the Lunar new year is an important occasion for Vietnamese people where the bonus of the 13th month's salary is paid to workers to support their new year celebrations [57]. Social entrepreneurs often sacrifice their own new year celebrations to support employees:

We never have any money lelf at lunar new year... All of our money is to pay for the workers, so we don't have any money at lunar new year. I am encouraged by the workers' recognition (Nhan).

Through such actions, respondents revealed resilience as a key quality of social entrepreneurs who are seeking to realize their destinies in meeting their cultural obligations as benevolent leaders who support others [58].

These entrepreneurs have developed organizations that fit with their cultural heritages and the contemporary socio-economic needs in society. The focus and function of their organizations reflect the collectivist values they espouse [18]. These values are transformed into the organization's philosophy and pro-social missions [15,23]. Devotion to their communities was the first philosophical tenet of participating social enterprises. The corresponding cultural value foundation for these enterprises reflects the entrepreneur's commitment toward a social mission as well as their inherent understandings of their community's needs. These understandings come from participants' standing as leaders within the communities they serve:

First of all, you have to understand your community. Frankly speaking, many people do start-ups that follow a trend, rather than truly understanding about the pain points of the community. I always advise them to study about the pain points of the community, deep in the pain of the community (Tú).

Participating entrepreneurs repeatedly reported targeting their enterprises to share the pain of the community and benefits of the enterprise: *"Your enterprise won't exist long if you*

*don't care about farmers' benefit"* (Hoa). This is an important consideration in a country that professes to become a 'Start-up Nation". However, it is also recognized by our participants that start-ups should not simply follow international trends. Enterprises should address actual community needs or 'pain points' so as to help as many people as possible.

This orientation towards community needs reflects core aspects of the traditional *village* culture that developed as a collective pillar of support across the upheavals of the nation's history [39]. In many respects, social enterprises are set up as village-styled and -led organizations. Central to village culture is the valuing of partnerships as a means of bringing people into harmony and the pursuit of shared endeavours. Partnerships not only support workers inside their enterprises, but also make broader contributions to the local region and sector:

> . . . We don't think that we are helping them. We are fair partners. They need to have their share in this partnership. The disabled people enterprise workers are treated as a regular partner as everybody (Thanh).

Participants were aware of the mutual benefits of partnerships and how foundational these are to the viability and success of social enterprises, and for amplifying any pro-social impacts:

> We prefer partners, rather than donors. Partners can contribute more to the development of our enterprise, bring more orders, create more jobs (Nhu).

Such statements reflect a spirit of self-reliance that is also associated with the village tradition [40] in that these entrepreneurs are not seeking charitable donations. They are seeking to weave their enterprises within various mutually supportive or harmonious partnerships. This openness extended to sharing experiences of success with other entrepreneurs as well as agencies trying to support social enterprise development in Vietnam.

Whilst being open to partnering with others, social entrepreneurs also emphasized the need for long-term cooperation: "*We need to build relationships that are long-term*" (Nhu). Such thinking has long been valued within traditional Vietnamese village culture [30,54] and participants emphasized the importance of finding, establishing and maintaining equitable partnerships:

> We need to have equality in partnership . . . We have to know how to listen . . . So that when we run our project, they will guide and help us out . . . . We need to be transparent with information. We need to dialogue with our people to reach common agreement . . . (Dia).

Dia emphasizes the need for openness, transparency and consultation when forging fruitful partnerships. This extract also reflects how a market-oriented economy has only existed in Vietnam for 30 years. Many enterprises are still forming and developing through consultations with Commune/village structures and government departments as part of a centrally planned economy [35].

In this section, we have considered the Viet worldview that emphasizes a collectivist response to addressing social problems through the development of social enterprises, which reflect the tradition of the village in Viet culture. Evident from participant accounts are important fits between village culture, the leadership of entrepreneurs, including their understandings of how fate brought them to social entrepreneurship, and the importance of cooperation, inclusion and partnerships both within and beyond the enterprises. Below, we further consider the organizational–societal fit through a focus on strategic leadership, village values and the pro-social performance missions for participating social enterprises.

### 3.2. Strategic Leadership, Inclusive Values and Pro-Social Performance

Employing many commercial business strategies, social enterprises utilize the revenue gained as a means of realizing pro-social missions [59]. These missions include developing and sustaining socio-economically inclusive community services that assist particular target groups (e.g., persons with disabilities), and address entrenched social and economic

issues through decent job creation [60]. This section considers how social entrepreneurs conceptualize and emphasize the importance of pro-social efficiency in their organizations in the context of the cultural values of inclusion and generosity as central to their social impact missions. The emphasis placed on pro-social efficiency departs from hegemonic global business models that associate efficiency with lean organizations. Within the collectivist Vietnamese cultural context [25], which values responsibilities towards others and socio-economic inclusion, efficiency is associated with supporting more employees who predominantly struggle to find jobs in the industrial sector.

Participants emphasized the importance of balancing the commercial and social impact factors of their organizations in order to ensure sustainability and realize their pro-social missions:

> For social enterprises, the 'enterprise' factor should be considered as much as the 'social' factor. We have to generate revenue and profit in order to pursuing our social mission. If we cannot generate revenue, we can't achieve the social mission. We gain customers by the product quality and design, not their sympathy (Tan).

Emphasis on growing the business was also associated with increasing the organization's ability to grow its reach in terms of supporting employees and target client groups. Emphasis was placed on the importance of having a good product that attracts more trade and enables the organization to realize economies of scale and financial surpluses that can be used in a range of ways to achieve pro-social missions:

> The first training course had 8 participants. They paid VND500,000 for the course . . . They shared with their colleagues, then the number of participants increased gradually, 15 people, 20 people, 30 people. At this moment, a training course during the school year is 120–150 people per course. In the summer holidays it is more than 200 people. 2 courses per month. Till now, the number of participants is 10,000, and our Facebook members are nearly 68,000 people (Thuỷ).

Pro-social efficiency is enabled once revenue has grown sufficiently. Tú also raised the importance of generating surplus resources from activities such as training workshops to cross-subsidize people who cannot afford to pay for such training, and to support job creation:

> There are students who are unable to pay the tuition free. However, we have free training for disadvantaged students on soft skills, job hunting skills. Our target is that each year we provide free training for at least 1000 students . . . Till 2019, we directly educated, trained, provided workshops, and job connection for 10.688 persons . . . In 2018, the number of jobs that we connected was 900.

In this extract, pro-social efficiency and additional positive social impacts are interwoven together within the efforts of the organization to subsidize access to training from students who can afford to pay to those who cannot afford to pay. The importance of growth or expansion was also presented—not in terms of profit taking, but in terms of supporting decent employment.

Contextual considerations are important here. As an emergent market, Vietnam has recently transitioned from an agricultural to an industrialized economy [61]. This has contributed to a reduction in absolute poverty for many but has also widened the gap between economically marginalized and affluent groups. Vulnerable groups, such as people living with disabilities and health concerns, often struggle to access new urban jobs in the industry [61]. Exercising responsive leadership, social entrepreneurs have embraced decent job creation and sustainability in their organizational missions. Generosity in employment creation manifests as a form of economic inclusion for marginalized groups, including middle-aged women who are often excluded from work in the new industrial zones:

> We hired quite a number of middle-aged women, aged 35 and over. There is no manufacturer in Mekong Delta industrial zone that hires them because

> factories only want young, easy-going people. My enterprise has flexible working mechanisms for these middle-aged women . . . I try to create a steady stream work for them, so they have a stable income according to their availability because they have to take care of their children . . . This is the social impact . . . My company does not use chemicals.... These women have changed from the toxic coconuts to my company, and their health is improved when working in my company. This brings happiness to them and their family (Sơn).

Social entrepreneurs repeatedly invoked the importance of increasing inclusion in healthy work environments that feature happy employees, good hygiene, were toxic free and that embraced flexible and positive working conditions.

With the development of industrial zones, the production of traditional handcrafts was under threat. One entrepreneur saw an opportunity to preserve the traditional craft industry by setting up a cooperative to employ people living with disabilities:

> The traditional handicraft was losing its market . . . Industrial zones are rising and attract young and healthy workers. My organization seeks to protect the traditional handicraft jobs and create jobs for poor, disabled and unhealthy workers... Although these people are not qualified to work in industrial zones, they still have high expenses, so they definitely need a job . . . There were seasons that I employed 500 disadvantaged people . . . Let me tell you a story. I had an opportunity to work on growing mushrooms business. If I grow mushrooms, I only need to employ 3–4 workers and can have profit of at least VND10,000,000/month. If I work on traditional handicraft, I employ hundreds of people with the same profit VND10,000,000/month. I choose to give up growing mushrooms... because of the hundreds of jobs that I can create from the same amount of profit instead of only 3 jobs (Nhân).

This entrepreneur demonstrates leadership by responding to the exclusion of vulnerable workers within the industrial zones. A central pro-social strategy is to maximize the number of jobs that are created through the handicraft enterprise, which also contributes to the quality of life of employees [62]. Behind such examples are tacit assumptions that derive from the collectivist mentality of the village and the pro-social valuing of those who have the means of helping others prioritizing such help over personal gains [30]. This orientation departs from the logic of the lean organization where efficiency and performance are associated with fewer jobs and increased profits [63,64]. For social entrepreneurs in this study, efficiency is equated with maximizing the number of sustainable employment opportunities [65,66].

In further interpreting this inclusive leadership orientation, it is important to consider the mutually supportive values of the traditional village, which emphasize the importance of leaders taking responsibility for the inclusion and livelihoods of as many beneficiaries as possible [40]. This orientation extends to cooperative centres with other persons beyond those directly employed in the enterprises (see farming example in the previous section). Central to such efforts are implicit assumptions about the importance of generosity of spirit in helping others not only through job creation, but also the quality of life of employees and their families, and access to nutritious food and healthcare [10]. Further, recent historical cooperatives formed under Communism appear to have also carried forward notions of collectivism, shared benefits and mutual support into the social enterprise space [35].

Whilst embracing generosity as a key facet in how these organizations operate and their self-appraisals of success, a necessary pragmatism is also apparent in participant accounts. That is, generous pay, work conditions and benefits are presented as examples of the 'right thing to do' to meet one's cultural obligations as a leader and associated pro-social mission to enhancing life in Vietnam for vulnerable groups. In this context, participants foregrounded the importance of fair pay and good, i.e., decent working conditions for improving both the mental and physical health of their employees. Emphasis was also placed on supporting employee satisfaction or happiness as this was seen as fostering

positive change in not only their lives, but also the lives of people living with them and the communities in which they dwell. These factors are also linked in turn to productivity gains that generate more income in a kind of virtuous circle:

> We see the improvement in their mental health. It is not only helpful for their work or themselves, but also for their beloved ones too. In return, they do their job better and that is good for my company. (Tấn)

When discussing these issues, participants routinely pointed to the importance of cultivating a harmonious and inclusive organizational culture, which can sustain happy and well-looked-after employees as a nuance of a positive work cycle [67]:

> A working environment where everybody is nice creates a positive energy. For example, whatever people are doing, they think about our social enterprise, they feel happy. Or when people partner with our social enterprise, in any circumstance, they feel positive energy making changes in their inner body, the huge inner change . . . When they are happy, they are committed with the organization. Their work is better. (Thao)

Creating such sustainable and productive livelihoods not only ensures the inclusion of marginalized groups in economic life, but can also lift them and their families out of poverty traps [68]. The consequences of such strategic leadership [66] extends beyond issues of fair pay to the provision of accommodation and food, and access to healthcare and related benefits. As Giai's proposes:

> We also offered free hearing support equipment for kids. Since we established in 2012 till now, I just wrote a report, we've given free more than 700 hearing support aids.

Such extracts reflect how social enterprises were providing significant positive impacts for not only their employees, but also their children, and members of the local community. Such strategic leadership also spills out beyond successful social enterprises through cooperative ventures with smaller organizations, including local farms that may be struggling in terms of viability. In reflecting further on the farm fertilizer example from the previous section, Thanh proposes that:

> The social impact that we've created is the contribution to agricultural community. The farmers experience the difficulties in production, but don't know how to deal with. We help them to deal with these difficulties, give them options to change while still maintaining effectiveness. We give them better solutions. They have positive change and have additional income monthly. They can use organic products instead of chemical ones. And they have additional income from their cooperation with us.

Thanh spoke to the importance of finding synergies and cooperative ventures with local farms as a means of increasing efficiencies for not only her enterprise, but also surrounding farms. Her organization collects farm waste to produce soil fertilizer that in turn can be used by the farms and increase their yields. Normally such waste from agricultural activities presents an additional cost for farmers because its disposal consumes considerable time and labour. However, Thanh now purchases such waste and also guides farmers in how to produce their own fertilizer from these by-products. This increases the revenue for all the enterprises involved. Such activities also exemplify how a pro-social orientation to enterprise has a way of broadening out to support the sustainability of the local ecology.

Exemplars such as the repurposing of farm waste also reflects how participating entrepreneurs value the cultivation of mutually beneficial fits between their own and their organizational values and local communities and associated enterprises [18,19,22]. An orientation towards pro-social efficiency can result in increased congruence between enterprise efficiency and generosity where commercial and social considerations are brought into harmony. Participating entrepreneurs took as a matter of course efforts to support local

socio-economic development in ways also consistent with the United Nations sustainable development goals [62]. As with social enterprises elsewhere, their strategic leadership often extended to acting as positive change agents within the broader community.

## 4. Conclusions

Culture is the lifeblood of any society. Correspondingly, to understand social enterprises in Vietnam, we have to understand the culture and how it shapes social entrepreneurship and leader's understandings of how they came to their pro-social work, the values enshrined in the organizations they create, their understandings of organizational efficiency and generosity and how they relate to external partners. In many respects, these enterprises comprise new articulations of traditional Vietnamese village culture where responsible leadership, studiousness, compassion, harmonization and inclusion are central to everyday socio-economic life [38]. Social Entrepreneurs establish, manage and lead their social enterprises with the spirit of solidarity and mutual support from this 'Village' root, creating uniqueness for the development of social enterprises in Vietnam. More importantly, with the belief in the natural order of the universe, chance and fate [69], social entrepreneurs turn the tensions between social and financial goals into creativity, novelty and sustainability.

The findings of this research support the assertion that societal context plays an important role in social enterprise development and leadership [13]. It is important to consider the cultural backgrounds and associated worldviews and enacted values of social entrepreneurs in order to extend our understanding of fit between the leaders, their organizations and the broader context, and in the present case, how cultural imperatives towards generosity shape social enterprise functions and missions [17]. Vietnamese social entrepreneurs are socialized within a collectivist culture and set of values that have been cultivated over centuries and that link the person, the village or community and the nation more broadly [29].

Our findings regarding the importance of Vietnamese traditional culture for the sustainability of social entrepreneurship and pro-social efficiency addresses a gap in context-orientated research into social enterprises [7]. This gap is only partially addressed by this exploratory study because we only engaged with the perspectives of founding entrepreneurial leaders. Future research is needed to explore employee perspectives and substantiate the benefits that are accrued to them and their families. However, the results of this qualitative study combined with the previous quantitative study [10] are proving useful in developing policy recommendations. We are currently in discussions with social enterprise sector organisations and the Vietnamese government regarding policy and related supports for enhancing the success and sustainability of social enterprise developments.

In closing, this paper foregrounds aspects of how this socio-cultural context has socialized social entrepreneurs to enact collectivist and harmonization values [30] and to construct their pro-social efforts as a key element of their destiny or fate in life. We have also highlighted and reformulated the Person–Environment fit theory within a non-Weird context and sector where it makes little sense to distinguish complementary and supplementary fits [69]. Fit with the Vietnamese 'villagerian' worldview and associated values comprise the backbone the links the social entrepreneur, social enterprises and pro-social changes that they contribute to society.

**Author Contributions:** Conceptualization, M.H.T.N., D.J.H. and S.C.C.; methodology, D.J.H. and M.H.T.N.; formal analysis, M.H.T.N. and D.J.H.; investigation, M.H.T.N. and D.J.H.; resources, D.J.H. and M.H.T.N.; data curation, M.H.T.N.; writing—original draft preparation, M.H.T.N.; writing—review and editing, D.J.H., M.H.T.N. and S.C.C.; visualization, M.H.T.N.; supervision, D.J.H.; project administration, M.H.T.N.; funding acquisition, S.C.C. and D.J.H. All authors have read and agreed to the published version of the manuscript.

**Funding:** This research was funded by MASSEY UNIVERSITY, GL16751252VIET.

**Institutional Review Board Statement:** The study was conducted according to the guidelines of the Declaration of Helsinki, and approved by the Ethics Committee of Massey University Ref NOR19/05 on 13 March 2019.

**Informed Consent Statement:** Informed consent was obtained from all subjects involved in the study.

**Data Availability Statement:** The data is not publicly sharable, since we specified in the MUHEC that it would stay anonymous and confidential, with password protection.

**Conflicts of Interest:** The authors declare no conflict of interest.

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
