# Peer review of "Fitting Social Enterprise for Sustainable Development in Vietnam"

_sustainability, doi:10.3390/su131910630_

Round 1
Reviewer 1 Report
Dear authors,
Thank you for submitting this manuscript. The title of the paper is interesting. It invites readers to become curios to find out how compatible social enterprises in Vietnam are to sustainable development. The aim of the paper is to reveal an understanding of the leadership of social enterprises and their impacts in relation to cultural and social considerations. Nevertheless, the paper is a scientific article, and several suggestions are relevant for its improvement.
Lines 14-24: The Abstract does not give a pertinent overview of the work. The abstract relates to the title, but it does not specify the main objective of the research. The abstract must place the question addressed in a broad context and highlight the purpose of the study. Is this purpose the importance of the leadership of social enterprises? Or the fit of social enterprises for the sustainable development? If it is about the latter, then what 1-4 enumeration represent? The fit of social enterprise and each of the four? Results and conclusions are missing.
Suggestions: Please reconsider the abstract section. Highlight the purpose of the study. Summarize the article's main findings. Indicate the main conclusions or interpretations.
Lines 186-195: The composition of the 20 social enterprises is clearly explained. Nevertheless, is the number of 20 social enterprises significant for the Vietnam economy? Which is the total number of enterprises? Why 20 and not 15 or 25?
Suggestions: Please explain the why the 20 number of social enterprises is relevant.
The authors paid little attention to the quality of the presentation. The description of the analysis and the results are not clear. There is still room for further explanations regarding the data and analysis.
Sections 3 and 4 – These sections include sentences and phrases from the interviews, but they look like a story and there is no clear structure of the text. What is the purpose of this story?
Suggestions: Please reconsider these sections in a structured manner, to help readers understand what the lessons are learned from the interviews. If you use tables or diagrams, figures or any other visual elements, the text will be clearer exposed. Do not forget where you start from (example – section 3 is about cultural ecosystem, and not all answers relate to this). Correct the sentences as not all of them are in commas.
Regarding the scientific soundness some recommendations should be considered.
Lines 204-209: The questions considered in the interviews are mentioned. Is this the full list of questions?
Lines 214-243: The interviews are described as semi-structured but the entire design of the interview analysis and interpretation is very vague. Was any interview guide developed or a pilot study conducted? Did you use any coding process? And subcodes?
Suggestions: If yes, please explain why this so general questions relate to the fit for sustainable development. If no, please mention all of them. Please describe the entire used method, with all the phases in a diagram format or a table to have the methodology proper explained.
Conclusion section. This section includes a description of the finding but this is related to the literature review and not on the own research. Limitations are missions. So are future research ideas.
Suggestion: Please reconsider the conclusion part and show the results from the own research. Add limitations of the study and future research directions.
Some English editing and major Journal editing rules are also important to be revised.
References in the text and in the list at the end of the manuscript do not follow the instructions of the journal.
Suggestion: Please check the: https://www.mdpi.com/journal/sustainability/instructions
Reviewer 2 Report
Well done research on Vietnam. Vietnam retains the characteristics of a traditional society before the Western influence in the future, and I could understand that this is a positive factor for the success of social enterprises.
At first, when the paper mentioned the role of leadership and the person-environment fit theory, I had a vague expectation that it could be classified into many different forms depending on the type of leadership and the type of fit. However, only one type related to Vietnam's social features is detailed in this paper. As Vietnam's social enterprises have unique characteristics (unlike western countries), I think the research deserves to be shared with academia as an introduction to social enterprises in Vietnam. Understanding Vietnam's cultural characteristics will also help in developing CSR-related plans in many third-world countries.
However, it is necessary to correct the references based on the journal format (a bit cumbersome to fix it).
Round 2
Reviewer 1 Report
Dear authors,
Thank you for the revised version of the manuscript. I appreciate the changes and the comments included in the cover letter.
Please pay attention to the references – numbering, embedding citation in the text, font and size.
Good luck in your future studies!
